# Population Genetics of California Gray Foxes Clarify Origins of the Island Fox

**DOI:** 10.3390/genes13101859

**Published:** 2022-10-14

**Authors:** Benjamin N. Sacks, Mark J. Statham, Laurel E. K. Serieys, Seth P. D. Riley

**Affiliations:** 1Mammalian Ecology and Conservation Unit, Center for Veterinary Genetics, Department of Population Health and Reproduction, University of California, Davis, CA 95616, USA; 2Panthera, 8 West 40th Street, 18th Floor, New York, NY 10018, USA; 3National Park Service, Santa Monica Mountains National Recreation Area, 401 W Hillcrest Dr, Thousand Oaks, CA 91360, USA

**Keywords:** gray fox, island fox, *Urocyon cinereoargenteus*, *Urocyon littoralis*

## Abstract

We used mitochondrial sequences and nuclear microsatellites to investigate population structure of gray foxes (*Urocyon cinereoargenteus*) and the evolutionary origins of the endemic island fox (*Urocyon littoralis*), which first appeared in the northern Channel Islands <13,000 years ago and in the southern Channel Islands <6000 years ago. It is unclear whether island foxes evolved directly from mainland gray foxes transported to the islands one or more times or from a now-extinct mainland population, already diverged from the gray fox. Our 345 mitochondrial sequences, combined with previous data, confirmed island foxes to be monophyletic, tracing to a most recent common ancestor approximately 85,000 years ago. Our rooted nuclear DNA tree additionally indicated genome-wide monophyly of island foxes relative to western gray foxes, although we detected admixture in northern island foxes from adjacent mainland gray foxes, consistent with some historical gene flow. Southern California gray foxes also bore a genetic signature of admixture and connectivity to a desert population, consistent with partial replacement by a late-Holocene range expansion. Using our outgroup analysis to root previous nuclear sequence-based trees indicated reciprocal monophyly of northern versus southern island foxes. Results were most consistent with island fox origins through multiple introductions from a now-extirpated mainland population.

## 1. Introduction

Many wide-ranging taxa evolved through alternating periods of range expansion in interglacial periods and range contractions into glacial refugia during glacial periods [1]. Glacial refugia typically were buffered from climatic extremes and, therefore, tended to harbor populations with the longest continuous histories and experienced high rates of speciation. The California Floristic Province (CFP) in western North America represents such a refugium [2]. The Pacific Crest mountain ranges (Sierra Nevada and Cascades) bound the eastern flank of the CFP and were glaciated during the late Pleistocene. This glaciated mountain chain served to retain the climate-stabilizing influence of the Pacific Ocean and presented barriers to animal movement, especially during glacial periods. Relative to the basins and deserts east of the Pacific Crest (hereafter, Desert ecoregion), which experienced large climatic oscillations during the Pleistocene, the stable climate and insularity of the CFP has resulted in preserving a more ancient record of ancestry in the DNA of some of its most vagile taxa [3,4,5].

The genus, *Urocyon*, represents a particularly enigmatic taxon within the CFP that includes two currently recognized species: the geographically widespread gray fox (*Urocyon cinereoargentus*) and the endemic island fox (*U. littoralis*). The island fox, which occurs only on the Channel Islands off the coast of Southern California and was named a distinct species primarily because of its smaller stature (~half-sized) and distinctive ecology, is thought to be of extremely recent origin [6]. Exactly how recent is unclear, but mitochondrial evidence indicated that it was no earlier than the Wisconsinan glacial period and as recent as the early Holocene [3]. Although *Urocyon* east of the Great Plains is currently regarded as conspecific with gray foxes west of the Great Plains, this grouping is paraphyletic with respect to island foxes; western and eastern gray foxes are characterized by deeply divergent (~1 MY), reciprocally monophyletic clades that are as divergent from one another as those corresponding to sister species [3,7]. Thus, despite the current taxonomy, from a strictly phylogenetic perspective, the eastern gray fox represents an outgroup relative to western gray and island foxes [3,7,8]. In this context, our interest in the current study was focused on western *Urocyon* (gray and island foxes).

The island fox is thought to have diverged from western gray foxes only about 13 kya, which is too recent to expect reciprocal monophyly between these named species [3,8,9,10,11,12]. Phylogenetic expectations for such a recent speciation event are incomplete lineage sorting (i.e., paraphyly) between gray and island foxes or for island foxes to exhibit monophyly consistent with a small founder population, in which case the most recent common ancestor would be no more than 13 ky divergent. Based on existing mitochondrial data, island fox matrilines form a monophyletic clade that has been estimated at 87–50 kya, which is tens of thousands of years before island foxes were thought to have inhabited the islands [3,7,8,9,10,11,12]. However, few gray foxes from Southern California have been sequenced, leaving open the possibility that additional sampling will identify mainland gray foxes also clustering in the island fox clade (i.e., negating its apparent monophyly). Alternatively, it is possible that the two species diverged on the mainland tens of thousands of years before island foxes were brought to the islands and before humans occurred on the landscape, as was previously hypothesized ([13,14,15], cited in [10]).

Evidence that island foxes were introduced to the northern Channel Islands by humans during the early Holocene <13,000 years ago is based primarily on paleontological and archaeological data, including radiocarbon dating of the oldest known specimen at 7600 BP [8,10,11,12,16]. Although foxes occur today on three northern islands (Santa Cruz, San Miguel, and Santa Rosa) separated from each other and the mainland by >10 km of open ocean, these islands formed a single super-island (Santarosae) prior to 9.5 kya. All known *Urocyon* specimens from the islands have borne the diminutive phenotype of extant island foxes. Thus, it remains unclear whether these original introductions involved gray foxes that rapidly evolved their small size on the islands or whether the island and gray foxes diverged prior to the island introductions [10,11]. Island fox populations of the three inhabited southern islands (San Nicolas, San Clemente, and Santa Catalina) were founded later, <6000 years ago [8,12,16]. In contrast to the northern islands, the southern islands were separated by >35 km from each other and the mainland. The source of introductions to the southern islands is traditionally presumed to be the northern islands, although mitochondrial data suggest that they more likely reflect three independent introductions from an as-of-yet unidentified population [3]. No fossils or faunal remains of any intermediate phenotype have been found on island or mainland, suggesting rapid evolution of the island phenotype at least once and possibly multiple times, or post-Pleistocene extirpation of a more divergent mainland form from which the islands were populated independently [12].

Our current understanding of the phylogenetics of gray and island foxes is primarily limited to inferences from mitochondrial data, which reflect only a single maternally inherited genealogy. Although several nuclear DNA studies of island foxes have explored relationships among island populations, few gray fox populations (i.e., from the mainland) were included in any of these studies, nor were outgroups included, thereby leaving it unknown whether island foxes exhibit nuclear genetic monophyly with respect to western gray foxes [17,18,19,20].

Here, we add new samples from a broad range of southern coastal California and east of the CFP to previous samples, cumulatively more than doubling those used for mitochondrial sequencing by Goddard et al. [3]; we use both mitochondrial and nuclear markers to investigate the antiquity of gray fox populations, their relationships to island foxes, and whether outgroup-rooted analyses of western *Urocyon* that include large numbers of gray foxes, particularly from Southern California, supports the mitochondrial and nuclear genetic monophyly of island foxes. Lastly, we describe the contemporary genetic structure of gray foxes throughout the CFP and the neighboring Desert ecoregion. Because males tend to be disproportionately responsible for gene flow among smaller canids, we anticipated higher gene flow reflected in nuclear markers than in maternally inherited mitochondrial data [21].

## 2. Materials and Methods

### 2.1. Samples

We used a total of 382 samples, including 166 previously published [3] and 216 new samples (Figure 1A; Appendix A). Samples included 329 western gray foxes primarily from California and Nevada, and, for reference, 31 eastern gray foxes from Georgia, USA. Because previous studies of island foxes thoroughly sampled the nuclear and mitochondrial genomes on all six inhabited islands and found them to be relatively homogeneous within island populations [18,19,20], we needed only a small sample from some of the islands to address our questions. In total, we used 22 island foxes mostly from the southern islands of San Clemente (*n* = 11), San Nicolas (*n* = 6), and Santa Catalina (*n* = 1), along with the northern island of Santa Cruz (*n* = 4). The 329 western gray fox samples were primarily from the CFP, including the North-Coast mountains (*n* = 83), Central Valley (*n* = 93), Cascades and northern Sierra Nevada (*n* = 47), Central Coast (*n* = 14), and Southern California (including two from northern Mexico, *n* = 51). The Southern California population was composed of two subpopulations, the Santa Monica Mountain population, which was directly adjacent to the northern Channel Islands, and the southern subpopulation, which occurred along the coast south of the Santa Monica Mountains and was adjacent to the southern Channel Islands (Figure 1B). Additionally, we obtained 41 samples from the Desert ecoregion, including the eastern slope of the Sierra Nevada Range, Great Basin Desert of Nevada, and a single sample from Yakima County, Washington. For spatial analyses of allele or haplotype frequencies of subsamples, we defined sampling sites of western gray foxes as centroids among geographically proximate specimens occurring in the same habitat ecoregion (Figure 1).

Samples included tissue (*n* = 47), blood (*n* = 110), mouth swabs (*n* = 4), and 221 noninvasively collected samples (*n* = 211 scats, *n* = 10 hair). Tissue samples were collected either post-mortem or as part of other studies from live individuals captured and released. Field collection procedures were approved by the University of California, Davis, Animal Care and Use Committee (IACUC No. 17860), the National Park Service Institutional Animal Care and Use Committee, and the California Department of Fish and Wildlife through scientific collecting permits and memoranda of understanding.

### 2.2. Laboratory Procedures

We extracted DNA from tissue using the DNeasy^®^ tissue kit (Qiagen Inc., Hilden, Germany) and from scats using the QiaAmp^®^ Stool Kit (Qiagen, Inc.) according to manufacturer’s instructions except that we eluted in 50 µL of buffer to concentrate DNA [22]. The mitochondrial DNA sequencing was as described previously [3], resulting in 785 bp (422 bp D loop, 363 bp cytochrome b). We accessioned all new haplotypes in GenBank (accession Nos. OP373713–OP373719). Ideally, genomic methods, such as restriction-site based reduced representation sequencing, could be used for our study. However, such methods require high-quantity and high-quality DNA and our samples, including many road-killed carcasses and noninvasive samples, were necessarily characterized by a wide range of DNA quality, most of which would be insufficient to allow such genomic methods. We, therefore, used 20 microsatellite loci in three PCR multiplex reactions (see Appendix A for primer sequences, concentrations, and dye sets). The DNA concentrations for blood and tissue extracts typically ranged 5–200 ng/µL and for fecal extracts, <1 ng/µL. Rather than quantitating DNA concentration for each sample, for efficiency and based on protocols developed for a wide range of similarly variable wildlife samples in our laboratory, we diluted all tissue extracts 100-fold prior to conducting PCRs and, for poor genotypes (e.g., allele peak heights smaller than standard peak heights), either reran undiluted or discarded. We also ran tissue-based PCRs separately from noninvasive DNA-based PCR, which were run in duplicate and combined into consensus sequences. Comparison of replicate genotypes indicated an average allelic dropout rate for noninvasive samples of 2.2% (SD = 4.4%). The PCR reactions were 10 µL (tissue DNA) or 11 µL (fecal/hair DNA), including 5 µL Qiagen multiplex master mix, 1 µL Q-solution, and primers at reaction concentrations ranging 0.25 to 1.00 µM (Appendix A), with 1 µL (tissue) or 2 µL (fecal/hair) of sample DNA. The thermal profile was 95 °C for 15 min, 33 cycles of 94 °C for 30 s, 58 °C for 90 s, and 72 °C for 60 s, followed by 72 °C for 10 min. We separated alleles using an ABI 3730 capillary sequencer in conjunction with Genescan 500 LIZ (Applied Biosystems) size standards and called peaks manually in STRand software (https://vgl.ucdavis.edu/STRand; accessed 30 April 2022).

### 2.3. Data Analysis

Based on local distributions of several mitochondrial haplogroups and their ages as estimated using rho statistics [23,24], Goddard et al. [3] demonstrated that gray foxes, sampled primarily from northern California, reflected relatively stable maternal ancestry long-pre-dating the last glacial maximum. Thus, our primary interest in adding mitochondrial data in this study from Southern California and the deserts to the east of the Pacific Crest was to examine relationships to island foxes, the stability of Southern California populations over this same time period, as well as the relationships of southern and northern California foxes and all of those in the CFP (i.e., both southern and northern California) to those east of the CFP. We constructed a haplotype network using the same 166 sequences used by Goddard et al. [3] along with the additional ones introduced in this study (Appendix A) and the median-joining algorithm [25] implemented in Networks (v 10.2.0.0). We weighted transversions 2:1 over transitions and weighted cytochrome-b substitutions 2:1 over D-loop substitutions. We rooted the network based on eastern gray foxes, and estimated ages of selected haplogroups in terms of the average number of substitutions separating ancestral and descendant nodes, or rho [23,24] assuming a mutation rate of 10.8% per million generations (2 years), equivalent to an expectation of one substitution every 11,795 generations (23,590 years) in the combined 785 bp cytochrome b and D loop sequence [3].

Prior to microsatellite analyses, we removed noninvasive samples (scats, hair) reflecting duplicates of the same individual and close relatives among all samples. Specifically, we assessed the pairwise relatedness within spatially defined samples (Figure 1) using a maximum likelihood estimator, implemented in ML Relate [26] and removed individual genotypes as necessary to ensure that no relatedness estimate between remaining individuals was greater than 0.375, which is the halfway point between that expected for second-order and first-order relatives. From remaining samples, we estimated observed and expected heterozygosity and average numbers of alleles per locus using microsatellite toolkit [27]. We estimated allelic richness, which adjusts for uneven sample sizes, and *F*_IS_ in FSTAT (Version 2.9.3.2; [28,29]). We tested for deviations in Hardy–Weinberg and gametic disequilibrium using permutation tests, followed by sequential Bonferroni corrections [30].

We used a Bayesian model-based approach implemented in program Structure v2.3.4 [31] to visualize geographic distributions of genetic clusters relative to bioregions [4,32]. We used the admixture model assuming correlated allele frequencies [33]. To assess likelihood associated with a range of predefined numbers of clusters (*K*), we initially conducted 10 runs at each of *K* = 1 to 8 clusters, with each run composed of 20,000 Markov chain Monte Carlo (MCMC) cycles, the first 10,000 cycles discarded as burn-in. We plotted average (and SD) logarithm of the probability of the data [LnP(D)] versus *K* as a guide to assist interpretations of cluster profiles at each level of *K*. We then conducted a final run at each of the best-supported levels of *K* consisting of 550,000 MCMC cycles (with the first 50,000 discarded), which was used for obtaining estimates of ancestry fractions (q).

To test the relative influences of isolation-by- (geographic) distance and discrete population structure, we also assessed partial correlations among pairwise distance matrices using a partial Mantel permutation test [34] in program Passage v 2.0 [35]. Specifically, we assessed whether Nei’s genetic distance (*D*_A_; [36]) was greater between sampling locations in different clusters than between sampling locations in the same cluster, while removing effects of geographic distance and vice versa. The independent variable matrices were of geographic (i.e., Euclidean) distance and cluster, which was defined in terms of the predominant cluster assignment of individuals within the sampling site. We used the cluster assignments from the admixture analysis described above at *K* = 3. The geographic distance matrix included distances measured around the San Francisco Bay and Delta estuary, which we assumed to present an absolute barrier to gene flow (e.g., [32]). In the cluster matrix, zeros indicated pairs of locations within the same cluster and ones indicated pairs of locations in different clusters.

To assess monophyly in island foxes, as well as other footprints of deeper phylogeographic structure, we estimated a neighbor-joining tree from the microsatellite data based on *D*_A_ with node support based on bootstrapping across loci. We rooted the tree to eastern gray foxes as an outgroup because other canids were too divergent (~10 MY) for microsatellite-based comparisons. Contrary to the current taxonomic recognition of the eastern gray fox as conspecific with the western gray fox and with the island fox as a distinct species, however, island foxes and western gray foxes represent a monophyletic group relative to the eastern gray fox (based on rooting to other canid genera [3,7,8]). The analysis was performed in program Populations [37].

## 3. Results

### 3.1. Mitochondrial

We obtained 179 new 785-bp mitochondrial sequences, which we added to 166 previously published sequences [3] to obtain 345 sequences in total (Appendix A). In addition to previously published haplotypes ([3]; GenBank accession Nos. KP888884-KP888897; KP888858-KP888883), we obtained several novel haplotypes (GenBank accession Nos. OP373713–OP373719). Despite the inclusion of 48 samples from Southern California (*n* = 46) and northern Mexico (*n* = 2), however, none had haplotypes falling within the island fox haplogroup (Figure 2). Most haplotypes from the Santa Monica Mountains (*n* = 28 of 33), along with those from the Central Coast (*n* = 9 of 14), grouped in the D-15 haplogroup, which we estimated to be approximately 15,000 years old (rho = 0.623, SD = 0.325). Except for the most basal haplotype (D-15), this haplogroup was not found anywhere else, suggesting that the matriline had been represented in the area since before island foxes were putatively introduced to the islands (i.e., <10,000 years ago; [8,12,16]). However, the D-15 haplotype itself occurred 50 km south of the Santa Monica Mountains. No other haplotypes from this haplogroup occurred in the 17 foxes sampled in the southern subpopulation. Thus, the distribution of haplotypes differed between the Santa Monica Mountains, nearest to the northern Channel Islands, and the southern subpopulation, closer to the southern Channel Islands. Rather than forming a closely related haplogroup, the southern subpopulation haplotypes reflected three unrelated haplogroups, consistent with admixture from range expansions. The D-15 haplotype was also common in northern California and the light gray haplogroup was common to the Desert ecoregion (Figure 2).

### 3.2. Microsatellites

We successfully genotyped 273 of the 380 samples at an average of 19.5 (SD = 1.2) microsatellite loci, including 119 of the 197 (60%) noninvasive samples. After excluding duplicates (*n* = 33) and close relatives (*n* = 47), we retained 193 genotypes for subsequent analyses, including 188 from Island foxes (*n* = 20) and western gray foxes (*n* = 168) and 5 from eastern gray foxes (Appendix A).

#### 3.2.1. Population Diversity

Measures of genetic diversity were higher for western gray fox population samples than for island foxes (Table 1). A slight and similar heterozygote deficiency (positive *F*_IS_) was evident in the 5 western gray fox population samples (significant in 4 of them), consistent with internal substructure. Most measures of diversity were highest in Southern California and the Desert ecoregion.

#### 3.2.2. Genetic Structure

The admixture analysis at *K* = 2–5 revealed hierarchical structure (Figure 3; Appendix A). At *K* = 2, gray foxes of the Desert ecoregion, Southern California, and the Central Coast, as well as Island foxes, all clustered together as distinct from gray foxes in northern California. At *K* = 3–5, Island foxes were differentiated from western gray foxes, although island foxes on Santa Cruz Island and, to a lesser extent, Santa Catalina Island, were assigned significant portions of their ancestry to the same cluster as Central Coastal and Southern California gray foxes. As *K* increased from 2 to 5, the gray fox cluster with which these island foxes were apportioned partial ancestry became more spatially limited, consistent with a closer relationship to the most geographically proximate mainland population. The relatively large extent of the Desert cluster regardless of *K* suggested high or recent connectivity throughout that ecoregion. This observation was consistent with the fossil record, which indicates an absence of gray foxes in that region until it was recently colonized through a post-Pleistocene population expansion [3]. At all levels of *K*, gray foxes spanning Southern California and the Desert ecoregion (eastern California, Nevada, and north to Washington) assigned to the same widespread genetic cluster. In contrast, 4 genetic clusters of western gray fox occurred within a relatively small portion of the CFP. Notably, at *K* = 5, microsatellites recovered a similar structure in Southern California as that suggested by the mitochondrial data. Specifically, the Santa Monica Mountains and Central Coast foxes clustered together whereas the remaining foxes of Southern California (i.e., southern subpopulation) were more admixed, sharing variable amounts of ancestry with the same cluster and with the Desert cluster (Figure 3B).

To better clarify relationships among island populations and with the mainland gray foxes, we performed a more local admixture analysis restricted to the 69 samples from the islands, Central Coast, and Southern California (both subpopulations), which further resolved relationships (Appendix A). At *K* = 3, the southern subpopulation was distinguished from the Santa Monica Mountain subpopulation and Central Coast population. At *K* = 4, island foxes were completely distinct from gray foxes, and northern and southern islands were distinct from each other except for the individual on Santa Catalina Island, which was admixed between Santa Cruz (northern) Island and the two other southern islands. This admixture of the Santa Catalina Island fox was consistent with previous findings suggesting recent (possibly in the past century) human introductions of foxes from Santa Cruz to Santa Catalina Island [3,8,17]. At *K* = 5, San Nicolas Island clustered as distinct. As with the corresponding analysis using all populations (Figure 3), the island foxes from the northern island (Santa Cruz) clustered most closely with Southern California gray foxes, but, in this analysis, those of the southern subpopulation.

Next, to further explore genetic structure of western gray foxes, we assessed continuity of gene flow using the full western gray fox data set. We observed a significant isolation-by-distance relationship (Mantel’s r = 0.69, *p* = 0.001), but the partial Mantel test (which controlled for geographic distance) nevertheless supported an additional effect of genetic cluster membership when tested at *K* = 3 (r_cluster-dist_ = 0.29; *p* = 0.04). This effect was relatively weak as reflected by the difference in elevations of the trend lines for isolation-by-distance within versus between clusters (Figure 4). The partial Mantel correlation between genetic distance and geographic distance, holding cluster distance (0,1) constant (r_geog-dist_ = 0.65; *p* = 0.001), was similar to the simple Mantel correlation that ignored cluster membership, suggesting that geographic distance accounted for most of the geographic structure.

A nuclear DNA population tree based on *D*_A_ estimated from the microsatellites and rooted to the eastern gray fox sample was largely concordant with structure inferred from the admixture analysis above. Specifically, the tree indicated high bootstrap support (97%) for reciprocally monophyletic clades corresponding to island foxes and western gray foxes (Figure 5). Among California gray foxes, the Cascades and northern Sierra Nevada (Pacific Crest), Central Valley, and North Coast clustered closely together distinct from those of Southern California and the Central Coast with high bootstrap support.

## 4. Discussion

Our analysis of mitochondrial DNA and nuclear genotypes of island and gray foxes yielded several important insights that contribute to our understanding of island fox origins and the antiquity and population structure of gray foxes within the CFP. First, by sampling approximately 50 additional gray foxes from coastal Southern California and Mexico nearest to the Channel Islands, we confirmed the presence of a large phylogenetic gap between island fox matrilines and those of the geographically most proximate gray foxes [3,8]. This observation supports the mitochondrial monophyly of island foxes with respect to modern gray foxes. Second, nuclear DNA also indicated that island fox populations formed a monophyletic group with respect to western gray foxes consistent with mitochondrial data. Although previous nuclear DNA studies had assumed monophyly among island foxes and rooted them to gray foxes from one or two western locations [17,18,19,20,38], our larger sample of gray foxes and inclusion of an outgroup (eastern gray foxes) to root the tree were necessary to confidently verify this putative monophyly. Third, our findings clarified the contemporary genetic structure of gray foxes throughout the CFP relative to those in the Desert ecoregion.

### 4.1. Origins of the Island Fox

By sampling a large number of gray foxes from Southern California, our results strengthen previous mitochondrial findings that island fox matrilines diverged from western gray fox matrilines long before the former were thought to have inhabited the islands [3,7]. Based on the estimated time to most recent common ancestor of island fox matrilines from whole mitogenomes, a minimum estimate of mitochondrial divergence between island and gray foxes would be 87,000 years [7], yet island foxes were presumably not present on any of the islands until 13–10 kya [12]. Additionally, our results suggest that at least some gray foxes of the mainland (Santa Monica Mountains) reflect continuous ancestry dating back to the time of the presumptive founding of island fox populations. Most of the phylogenetically distinct haplotypes we observed in the mainland location nearest the northern Channel Islands, the Santa Monica Mountains, formed an endemic D-15 haplogroup that dated to nearly 15,000 years before present. Because mitochondrial data can only indicate a single genealogy, we do not necessarily consider the observation of different single surviving matrilines on the mainland versus the northern islands to strongly argue against their having been derived from the same ancestral pool ~13 kya. Rather, the observation serves to clarify that both populations can be traced back to their current locations since that time. To better understand the relationship between these populations, therefore, it was necessary to examine nuclear DNA patterns. Consistent with mitochondrial findings, the dichotomous tree and admixture analysis at *K* > 2 also implied the predominance of common nuclear-genetic ancestry unique to the genomes of all island foxes. Thus, both mitochondrial and nuclear DNA highlight that the predominant ancestry of island foxes was not represented in the contemporary mainland gray foxes we sampled.

The significance of the missing link between island and nearby mainland gray foxes is magnified by the long span of time over which northern and southern islands were colonized. The archaeological, cultural, and DNA evidence all agree in suggesting that foxes arrived on the northern Channel Islands approximately 13–7500 kya, long before they arrived in the southern islands, closer to 5.5 kya [3,8,10,11,12,16,17]. The mitochondrial pattern of the northern islands is similar to that of the island spotted skunk (*Spilogale gracilis amphialus*), which also inhabits two of the northern islands and was thought to have arrived around the same time as foxes [39]. Both species exhibit a single mitochondrial haplogroup endemic to the northern islands. In both cases, distributions of haplotypes and their nucleotide diversity were consistent with origins from a single maternal founder during the time sea level was low enough for the northern islands to form the single super-island, Santarosae, up to slightly <10 kya [3,39]. Thus, other than the missing mitochondrial link between the northern island haplogroup and modern gray foxes, there are no major conflicts in the evidence for the timing of colonization in the north. The bigger mystery pertains to the southern islands, which were founded much later in the Holocene. In addition to the fossil/sub-fossil evidence suggesting no foxes inhabited the southern island prior to 6000 years ago at earliest [8,12,16], a recent study that reconstructed historical population sizes based on whole genome resequencing of island foxes on two of the southern islands (Santa Catalina, San Clemente) found a precipitous decline consistent with a founder effect in these populations during the putative time of their founding [40].

The hypothesis traditionally put forth to explain the current distribution of island foxes posits that they evolved their small body size once after colonizing or being transported to the northern islands when they formed a single super-island, Santarosae, during the late Pleistocene (approximately 13,000 years ago), and were subsequently spread by humans to successive southern islands in mid-Holocene [8,9,10,17]. This model is partly based on parsimony, assuming the island phenotype evolved only once, and conforms well to archaeological and fossil/subfossil data, which primarily support the hypothesized timing. Contrary to this model, however, our findings combined with previous genetic data support an early hypothesis that fell out of favor decades ago, specifically, that island foxes derive from a now-extinct mainland population ([13,14,15], cited in [10]).

As initially argued by Goddard et al. [3] on the basis of mitogenomes [8], foxes on the southern islands were likely derived independently from one another and from those on the northern islands, most likely by separate introductions from the mainland. Among island populations, both types of DNA indicate reciprocally monophyletic northern and southern subclades. First, by inferring the root based on our nuclear DNA tree, previous whole genome and reduced representation sequencing data (as well as microsatellites) imply reciprocally monophyletic subclades distinguishing southern from northern island populations [18,19,20,38]. Second, a Bayesian phylogenetic tree based on whole mitogenomes of island and gray foxes rooted to eastern gray foxes (and *Vulpes* and *Canis* spp.), showed reciprocally monophyletic mitochondrial subclades between northern versus southern islands [7]. If the southern island populations had been derived initially from transportation of foxes from a northern island, either to all southern islands or from successive movements between southern islands, the topological expectation (particularly from nuclear DNA) would be for southern island populations to be nested within that corresponding to the northern island that sourced them, with other northern island populations basal to these.

The reciprocally monophyletic northern and southern subclades are therefore more consistent with the occurrence of at least two distinct founding events. The mitochondrial data further suggest that each of the three southern islands was sourced from a different founder (analysis of [8] mitogenomes by [3]). The northern island mitochondrial haplogroup contained 9 distinct haplotypes, all within the same haplogroup estimated to be approximately 13,500 years old (SD = 3100 years), whereas the entire island fox mitochondrial clade was estimated to be 51,000 [3] to 87,000 [7] years old depending on calibrations. Each of the southern islands contained a haplogroup that was >10,000 years divergent from any other. Considering the existing evidence that no foxes occurred on these southern islands before the mid-Holocene [8,10,11,12,16], separate introductions seems the most plausible explanation. Given that all 9 haplotypes in the northern islands form a single endemic matriline, it seems implausible that the three distinct matrilines of the southern islands could be derived from the northern islands, leaving the mainland as the most likely source. Thus, an important question is how four divergent matrilines of the island fox clade supplied the islands over several thousand years, yet were unrepresented in any modern gray foxes from the nearby mainland.

The most parsimonious explanation seems to be that the direct maternal ancestors of all island foxes occurred on the Southern California mainland during the time span over which foxes were established on the islands, i.e., 13,000–5500 years ago, but subsequently disappeared, possibly through replacement by different populations of *Urocyon* that expanded from elsewhere during the late Holocene (Figure 6). Our data were consistent with such a scenario. Most importantly, our nuclear DNA admixture analyses indicated that some island foxes contained ancestry that clustered most closely with the Southern California and Central Coast populations as expected if island foxes were sourced from one of those nearby mainland locations. Although we found no evidence that the Santa Monica Mountain population reflected recent turnover of gray foxes as would be necessary to explain such a source, those south and east of the Santa Monica Mountains (southern subpopulation) differed starkly in their genetic composition. Both nuclear and mitochondrial data indicated high admixture within the southern subpopulation and high connectivity to the Desert ecoregion, itself known to be colonized as a result of post-Pleistocene expansions [3]. Comparison of haplotypes in the southern subpopulation with those from east of the Pacific Crest, including those previously published from as far east as Texas [7] (Appendix A), suggest that they originated from multiple sources, consistent with immigration related to post-Pleistocene expansions from the north, east, and south.

If, as our data suggest, the population ancestral to island foxes was extirpated from the mainland within the past several thousand years, it remains possible that they were more phenotypically similar to modern island foxes than gray foxes, potentially reflecting a speciation event considerably older than the timing of their introduction to the islands ([13,14,15], cited in [10]). If so, such a phenotypic distinction between contemporary populations could help to explain why none of the haplogroups reflected in today’s larger-bodied mainland gray foxes were represented on any of the islands.

### 4.2. Antiquity and Genetic Structure of California Gray Foxes

Our mitochondrial and nuclear DNA findings were largely concordant in showing two distinct gray fox populations in the CFP. In particular, northern California, composed of the North Coast, Pacific Crest (Cascade and western Sierra Nevada), and Central Valley samples, stood out as harboring the most ancient record of ancestry, including an endemic haplogroup (Red) estimated to be over 70,000 years old. Correspondingly, the microsatellite tree and the admixture analysis distinguished northern California from all other western *Urocyon* populations, even including the island foxes at *K* = 2. The relative antiquity of the northern California populations likely relates to their insularity during the late Pleistocene. The Pacific Crest served both as a barrier to movement as well as to render the climate to the east inhospitable to gray foxes during glacial periods, thereby isolating gray foxes within the northern portion of the CFP from all other refugial populations except those to the south.

On a more contemporary timescale, both admixture and isolation-by-distance analyses indicated little nuclear-genetic evidence of discrete population structure within the CFP, suggesting that gene flow was relatively continuous. Most of the genetic differentiation among subsamples could be explained by geographic distance, although discrete clusters identified in the admixture analyses contributed slightly. Aside from geographic distance, the pattern of habitat on the California landscape likely determined genetic connectivity. Gray foxes currently occupy a wide range throughout California and tend to be most abundant in scrublands or forests with dense understories, but rarely occur at high elevations (>2400 m) or in open agricultural or desert terrain [41,42]. Consequently, much of the Pacific Crest along the Cascades and Sierra Nevada Ranges likely presents a barrier to movement. Additionally, the Central Valley is largely devoid of gray foxes or suitable habitat except where remnant riparian water-courses extend forested habitats into the valley or in various islands of scrub habitat, primarily at the northern end of the state [43]. Both mitochondrial and microsatellite data indicated some genetic connectivity between northern California and the Desert ecoregion at latitudes consistent with gene flow over the lower-elevation passes of the northern Sierra Nevada Mountains. We lacked samples from the southern end of the western Sierra Nevada Range and the Tehachapi Range, but these regions represent a narrow isthmus of habitat potentially linking northern and southern California populations.

## 5. Conclusions

Our findings suggest that the mainland ancestors of the island fox were most likely extirpated and replaced by gray foxes expanding from multiple locations during the late Holocene. Although we found only a weak genetic connection between island foxes and gray foxes, its correspondence to the nearest mainland location supports the existence of at least some gene flow in the past between ancestors of island and gray foxes. Nevertheless, the absence of the primary source of ancestry for the island fox on the mainland leaves open the possibility that this extirpated ancestral population could have attained the smaller phenotype prior to its transportation to the islands. Additional archaeological data from the coastline of Southern California would be useful in answering this question. Although ours was the first nuclear DNA study to include a large sample of western gray foxes and an outgroup, along with island foxes, microsatellites represent a crude tool. In the future, use of more powerful genomic approaches would lead to greater insights into the enigmatic recent evolutionary history of the island fox and its relationship to gray foxes. Addition of ancient DNA in conjunction with morphological data would be especially valuable in clarifying the species-level divergence between island and gray foxes.

## Figures and Tables

**Figure 1 genes-13-01859-f001:**
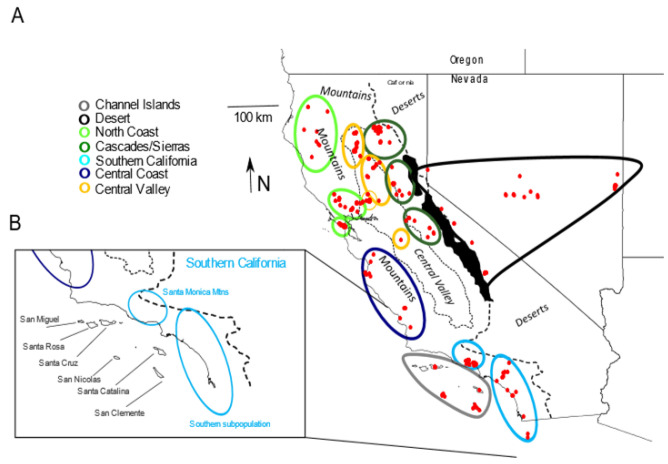
(**A**). Map of 382 western gray fox (*Urocyon cinereoargenteus*) and island fox (*U. littoralis*) samples (red dots) with sampling sites defined by ellipses. Colors of ellipses indicate the broader regional samples used in population tree and genetic diversity statistics. Island fox samples are shown on the four sampled Channel Islands. Black-filled polygons indicate previously glaciated portions of the Sierra Nevada Range. Not shown are a single sample from Yakima County, Washington (Desert sample) and samples from the eastern gray fox in Georgia. (**B**). Inset: names of the six fox-inhabited Channel Islands and two subpopulations of mainland Southern California: the Santa Monica Mountains (including Simi Hills) and the southern subpopulation.

**Figure 2 genes-13-01859-f002:**
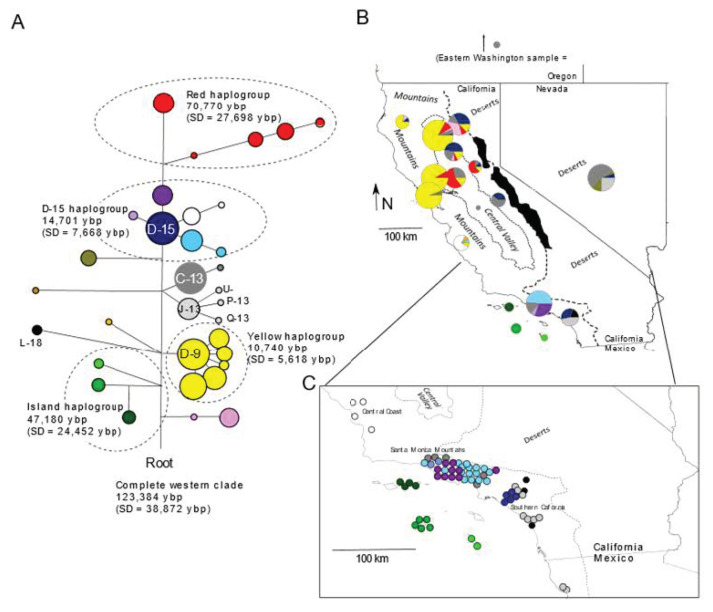
Mitochondrial diversity in gray foxes (*Urocyon cinereoargenteus*) and island foxes (*U. littoralis*) in the California Floristic Province and the Desert ecoregion separated by the Pacific Crest mountains. (**A**). a median-joining network composed of 785 bp of cytochrome b and D loop sequence, illustrating several haplogroups and their estimated ages based on rho statistics and a substitution rate of 10.8% per million generations (5.4%/MY); branch lengths are proportional to the number of substitutions and node sizes are proportional to the number of samples; red and yellow (D-9) haplogroups are endemic to northern California, whereas the D-15 haplogroup, except for the basal haplotype D-15 itself, are endemic to Southern California and Central Coast; the island fox haplogroup is endemic to the Channel Islands. (**B**). a map of haplotype pie charts corresponding to the 15 subsamples, with colors corresponding to nodes in panel A; black-filled polygons along the Pacific Crest indicate the location of glaciers during the Pleistocene epoch. (**C**). enlarged portion of Southern California, illustrating individual locations jittered for visibility and color-coded according to the nodes in panel A; two samples from Mexico were based solely on cytochrome-b haplotypes (U).

**Figure 3 genes-13-01859-f003:**
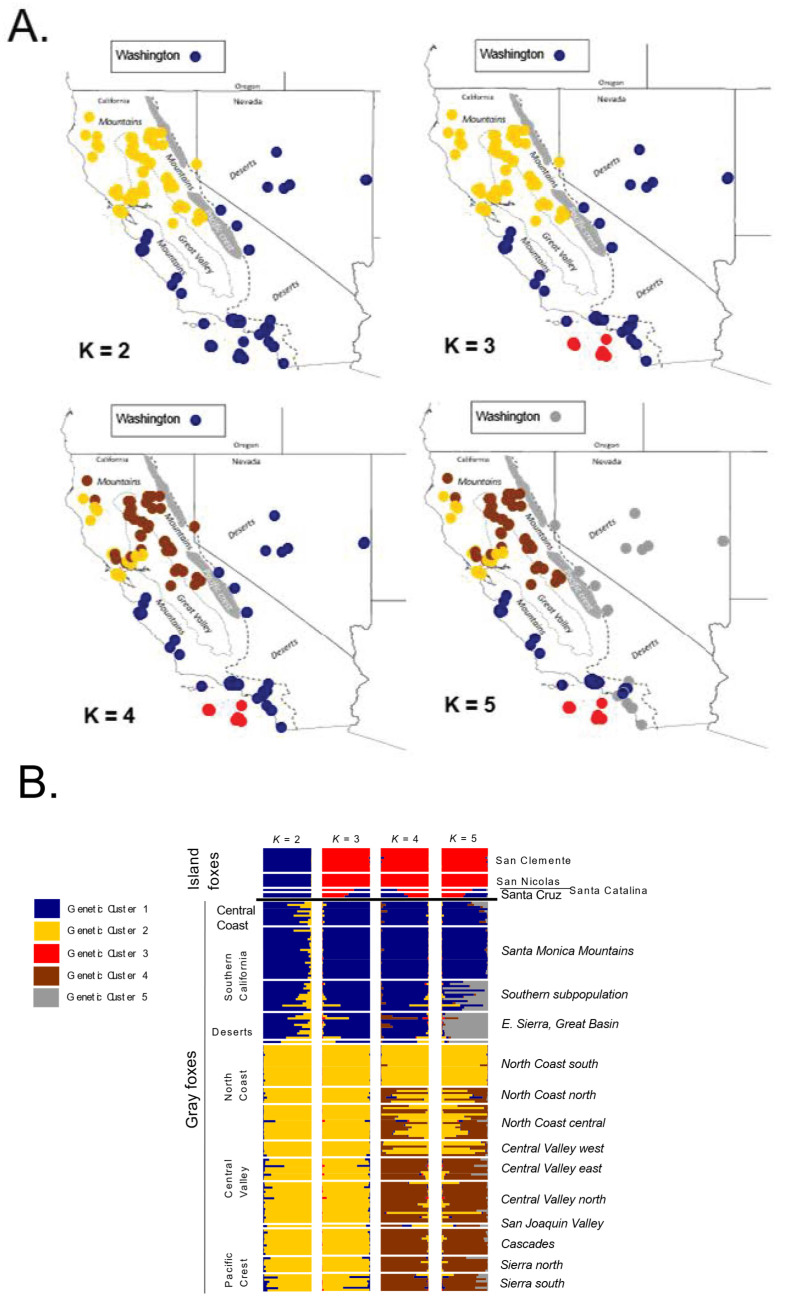
Admixture analysis of 188 western gray foxes (*Urocyon cinereoargenteus*) and island foxes (*U. littoralis*) with 20 microsatellite loci conducted in program Structure, assuming *K* = 2, 3, 4, and 5 clusters. *(***A**). Sample locations represented on maps as dots colored corresponding to the genetic cluster with the highest *q*-value. *(***B**). Horizontal bar charts showing *q* values for all clusters assigned to each fox, arranged by species (island fox, top; gray fox bottom) and gray fox population (left) and subpopulation (right). Log probability of the data (SD) for *K* = 1–10 were as follows: −11,828 (1.59), −10,943 (2.1), −10,268 (7.71), −10,047 (2.04), −10,010 (14.48), −9982 (24.76), −9896 (69.98), −9976 (96.81), −9944 (204.73), and −10,320 (556.45).

**Figure 4 genes-13-01859-f004:**
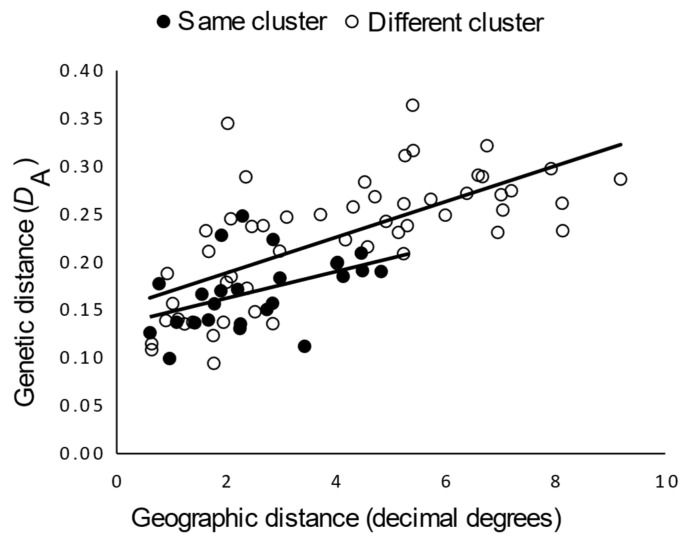
Relationship between Nei’s genetic distance and geographic distance between gray fox (*Urocyon cinereoargenteus*) sampling sites within vs. between genetic clusters (*K* = 3). Nei’s genetic distance (*D*_A_) was estimated from 20 microsatellite loci for each pair of sampling sites (Figure 1), which are indicated as comparisons between sampling sites assigned to the same genetic cluster (open circle) and to different genetic clusters (filled circles). Genetic clusters were defined for each sampling site as the one for which the greatest ancestry (*q* value) was assigned in the admixture analysis (see Figure 3). Trend lines illustrate statistically significant relationships based on partial Mantel tests.

**Figure 5 genes-13-01859-f005:**
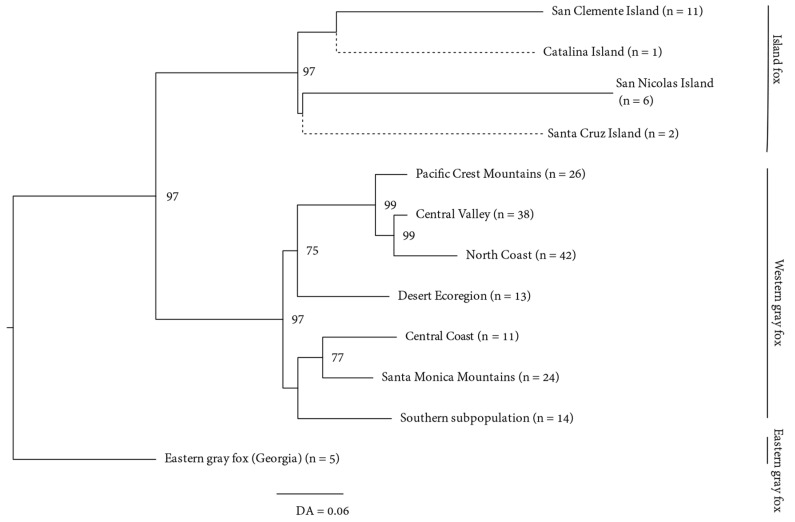
Neighbor-joining population tree illustrating reciprocal monophyly between a group of 4 Island fox (*Urocyon littoralis*) populations versus 7 western gray fox (*U. cinereoargenteus*) populations, rooted to an eastern gray fox population outgroup. Branch lengths are based on Nei’s genetic distance (*D*_A_) [36] calculated from 20 microsatellite loci and 999 bootstrap replicates. Only bootstrap values >65% are shown. Bootstrap values were calculated without Santa Catalina and Santa Cruz island populations included because of low sample size.

**Figure 6 genes-13-01859-f006:**
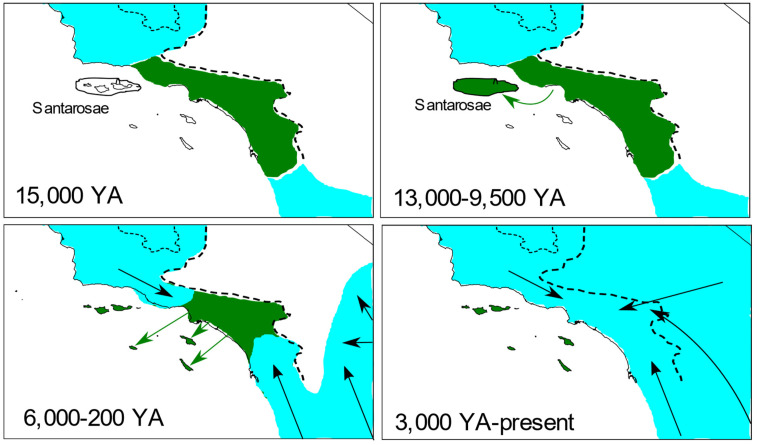
Schematic representation of a hypothesis explaining the origins of island foxes (*Urocyon littoralis*) from a mainland ancestor replaced by expanding populations of gray foxes (*U. cinereoargenteus*). Island foxes and their immediate mainland ancestors (green) are distinguished from gray foxes (blue) 15,000 years ago (YA), prior to their introduction to the northern super-island, Santarosae 13,000–9500 YA, followed by their independent introductions to each of the southern islands 6000–200 YA, and finally their extirpation and replacement by expanding gray foxes on the mainland during the late Holocene. Overlapping time intervals reflect uncertainty in timing of extirpation and replacement. Putative introductions of foxes to the Channel Islands are indicated by green arrows and putative expansion routes of gray foxes are indicated by black arrows. Dashed lines indicate boundary to the California Floristic Province and dotted line bounds the Central Valley.

**Table 1 genes-13-01859-t001:** Population genetic statistics based on 20 microsatellite loci for 188 island foxes (*Urocyon littoralis*) and western gray foxes (*U. cinereoargenteus*) from 4 islands and 5 mainland sampling locations in western North America, including expected heterozygosity (*H***_e_**; under Hardy–Weinberg expectations), observed heterozygosity (*H***_o_**), average numbers of alleles per locus (N_A_), allelic richness (A_R_), and inbreeding coefficient (*F*_IS_).

Sample	*n*	*H*_e_ (SE)	*H*_o_ (SE)	N_A_ (SE)	A_R_ (SE)	*F*_IS_ (SE)
Island foxes (all) ^1^	20	0.43 (0.07)	0.15 (0.02)	3.15 (0.36)	2.21 (0.19)	0.59 (0.08) *
*S. Clemente Island*	11	0.27 (0.06)	0.19 (0.03)	2 (0.21)		0.29 (0.09) *
*S. Nicolas Island*	6	0.04 (0.02)	0.04 (0.02)	1.16 (0.09)		−0.14 (0.1)
Deserts	13	0.70 (0.06)	0.58 (0.03)	6.6 (0.63)	3.51 (0.25)	0.17 (0.04) *
North Coast	42	0.61 (0.05)	0.52 (0.02)	5.7 (0.65)	2.89 (0.22)	0.17 (0.04) *
Cascades/Sierras	26	0.61 (0.06)	0.57 (0.02)	6.4 (0.70)	3.04 (0.26)	0.07 (0.03)
Southern California (all)	38	0.68 (0.05)	0.58 (0.02)	6.9 (0.63)	3.32 (0.22)	0.17 (0.05) *
*Santa Monica Mtns*	24	0.64 (0.05)	0.55 (0.02)	5.6 (0.50)		0.13 (0.05) *
*Southern subpopulation*	14	0.70 (0.05)	0.61 (0.03)	6.2 (0.73)		0.12 (0.05) *
Central Coast	11	0.63 (0.06)	0.62 (0.03)	5.1 (0.48)	3.16 (0.24)	0.02 (0.05)
Central Valley	38	0.63 (0.05)	0.56 (0.02)	6.7 (0.67)	3.07 (0.24)	0.12 (0.03) *

^1^ Island foxes (all) include 17 from the two islands shown in italics as well as 3 more from Santa Catalina (*n* = 1) and Santa Cruz (*n* = 2) Islands. * *F*_IS_ significantly different than zero at *p* < 0.05.

## Data Availability

Data are contained within the article and Appendix A; new sequences were additionally accessioned in GenBank (Nos. OP373713–OP373719).

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
