# Peer review of "Population Genetics of California Gray Foxes Clarify Origins of the Island Fox"

_genes, 2022, doi:10.3390/genes13101859_

Round 1

Reviewer 1 Report

In the manuscript "Population genetics of California gray foxes clarify origins of the island fox," Sacks et al. used mitochondrial and microsatellite data to investigate the population structure of grey foxes and the origins of the island fox. The manuscript is very well written and easy to understand. Yet, in my opinion, some of the analyses need major improvements or a more detailed explanation of why they were done the way they were done. 

Major comments: 

One of my main concerns is regarding the phylogenetic analyses. IF island foxes are a distinct species, how can they nest within the gray fox populations?  OR, why was the eastern gray fox population chosen as an outgroup when in fact, it is considered the same species as the western gray fox? For me, that clearly violates the assumption of an outgroup, and I would suggest including a different species that is unequivocally an outgroup, e.g., the red fox, as the root. 

Furthermore, if the island fox is monophyletic but groups with the western gray fox population and if the island fox is considered a distinct species, the authors need to discuss the species status and systematics of the eastern and western gray foxes, as they would clearly be paraphyletic. It is not mentioned at all that Eastern and western gray foxes are considered subspecies. Rather than speculating about the origin of island foxes for multiple pages, I would suggest that the authors put some more effort into discussing the taxonomy and systematics of these taxa. If others have done so, why did the authors not mention it at all in the introduction? I am sorry to put so much emphasis on this issue, and I am sure that the authors wanted to avoid a taxonomic discussion. However, the presented data raises this question, and in my opinion, it needs to be discussed properly. 

Minor comments: 

L44-47: Please explain on what basis the island foxes are considered a distinct species. Just the size difference and geographic origins?

L137-142: I would prefer that more details are given in the main manuscript about the dye set and platform used for fragment analyses.  In addition, why was the DNA concentration not measured prior to PCR, and was the template volume adjusted based on the concentration?

L173: not sure if that was just an error, but the header "laboratory procedures" would fit much better before the DNA extraction and PCR section, and from the numbering, it seems that was also the intention.

Figure 3A,B: colors do not match at K=2

Reviewer 2 Report

In this manuscript, the authors analyzed mitochondrial and nuclear sequences to investigate the origins of the endemic island fox on the different Channel Islands and their ancestral connection to gray foxes in Southern California. The DNA data was also used to explore the population structures of the gray foxes in the California Floristic Province and neighboring desserts.

The authors gathered additional samples and used nuclear satellite sequences in addition to the mitochondrial marker that were used in previous studies. This paper delivers a new and interesting analysis of the population history of the two species, Urocyon littoralis and Urocyon cinereoargenteus

However, the study has several issues as stated in the major points:

1)    In Material and Methods, please add the concentrations of the DNA that were used for the PCRs (lines 137-142). The used volume without DNA concentration is not very useful.

2)    Figure 1 is very difficult to understand. What are the circles on the left referring to? It seems that there is nowhere a red ellipse. What is the meaning of the red strokes? The colors of the ellipses are difficult to decipher because they are too thin. It would be helpful to use more labelling such as a,b,c … for the subpanels to have an organized legend that I much easier to follow.

3)    Labelling of the subpanels in Figure 2 would be helpful too. Describing panels in the legends with left or upper right seems unprofessional. In addition, the legend jumps between the panels which makes it hard for the reader to follow. It would help to describe the network on the left in more detail.

4)    The details in Figure 3 A are too small and therefore impossible to decipher. Please add a color code for Figure 3B.

5)    Figure 5 needs additional information in the legends. What is exactly plotted? The reader should be able to understand the Figure with the legend without reading the text and/or Material and Methods.

6)    The discussion is very long and difficult to follow. Please reorganize and shorten it. A schematic would help the reader to understand the discussion and conclusions.

Round 2

Reviewer 1 Report

I want to thank the authors for the quick and detailed response. Many of my concerns and questions have been addressed; however, I am still not convinced about the msat phylogeny. The sister taxon relationship between western gray foxes + Island foxes, and eastern gray foxes, was, as far as I could see, only established based on mtDNA. We have seen in many taxa that the relationships can look strikingly different on the nuclear level due to processes such as mitochondrial capture. 

I acknowledge the issue with large divergence times to other, for me better-suited, outgroup species, yet, this only shows that microsatellites were likely not the right marker system to choose. Alternatives with more power to resolve this issue, are available for many years, and I am honestly surprised that the authors still choose to use microsatellites over methods such as, e.g., ddRAD.  

Even though the authors did add more information about the microsatellite analyses into the methods, much is still missing or questionable. The authors say that they did not measure DNA concentration but just diluted them all by 1:100. This might have worked for many samples but is not really a standard way of doing it. Also, how did they deal with samples that did not amplify or did not show peaks? Were they repeated? Was the DNA concentration checked fo them and repeated with more template DNA? Have samples really been amplified only once or twice (for fecal samples)? In my opinion, msats always require at least three replicates to evaluate the accuracy of the genotypes properly. 

I am sorry to say that with the additional information the authors added to the methods, I am even less convinced that this study should be published in its current form. The study design seems to me quite flawed. I am also not sure why the authors focus only on the origin of the island foxes, when there is clearly a much larger and more pressing issue at hand. For me, it would have made way more sense to study the taxonomic structure (incl. species, subspecies, and populations) of Urocyon and, within this study, tackle the origins of island foxes.  

Round 3

Reviewer 1 Report

I thank the authors for the very quick response to my criticism, and even if I do not agree with some of their explanations, the authors have convinced me enough to publish this manuscript.  

One last thing that I would like to add, however, is that more samples, in general, are better, yet, sometimes fewer samples, but more information per sample in the form of a few thousand SNPs might result in deeper insights and a better understanding of the species/taxon. 

I hope that the authors did see my comments as constructive and not as unfair, as this was not my intention, and I am looking forward to reading more about this interesting taxon from the authors in the future.